# Accuracy and Clinical Impact of Estimating Low-Density Lipoprotein-Cholesterol at High and Low Levels by Different Equations

**DOI:** 10.3390/biomedicines10123156

**Published:** 2022-12-06

**Authors:** Maureen Sampson, Anna Wolska, Justine Cole, Rafael Zubirán, James D. Otvos, Jeff W. Meeusen, Leslie J. Donato, Allan S. Jaffe, Alan T. Remaley

**Affiliations:** 1Department of Laboratory Medicine, Clinical Center, National Institutes of Health, Bethesda, MD 20892, USA; 2Lipoprotein Metabolism Laboratory, Translational Vascular Medicine Branch, National Heart, Lung and Blood Institute, National Institutes of Health, Bethesda, MD 20892, USA; 3Salvador Zubirán National Institute of Health Sciences and Nutrition, Mexico City 14080, Mexico; 4Department of Laboratory Medicine and Pathology, Mayo Clinic, Rochester, MN 55902, USA

**Keywords:** low-density lipoproteins, cholesterol, triglyceride, cardiovascular disease risk

## Abstract

New more effective lipid-lowering therapies have made it important to accurately determine Low-density lipoprotein-cholesterol (LDL-C) at both high and low levels. LDL-C was measured by the β-quantification reference method (BQ) (N = 40,346) and compared to Friedewald (F-LDL-C), Martin (M-LDL-C), extended Martin (eM-LDL-C) and Sampson (S-LDL-C) equations by regression analysis, error-grid analysis, and concordance with the BQ method for classification into different LDL-C treatment intervals. For triglycerides (TG) < 175 mg/dL, the four LDL-C equations yielded similarly accurate results, but for TG between 175 and 800 mg/dL, the S-LDL-C equation when compared to the BQ method had a lower mean absolute difference (mg/dL) (MAD = 10.66) than F-LDL-C (MAD = 13.09), M-LDL-C (MAD = 13.16) or eM-LDL-C (MAD = 12.70) equations. By error-grid analysis, the S-LDL-C equation for TG > 400 mg/dL not only had the least analytical errors but also the lowest frequency of clinically relevant errors at the low (<70 mg/dL) and high (>190 mg/dL) LDL-C cut-points (S-LDL-C: 13.5%, F-LDL-C: 23.0%, M-LDL-C: 20.5%) and eM-LDL-C: 20.0%) equations. The S-LDL-C equation also had the best overall concordance to the BQ reference method for classifying patients into different LDL-C treatment intervals. The S-LDL-C equation is both more analytically accurate than alternative equations and results in less clinically relevant errors at high and low LDL-C levels.

## 1. Introduction

Cholesterol in low-density lipoproteins (LDL) (density range: 1.006–1063 g/mL), is causally related to the development of atherosclerosis [1]. Although other biomarkers for risk stratification such as apolipoprotein B (apoB) may be superior [2,3], the accurate measurement of LDL cholesterol (LDL-C) at both low and high levels is still important when following current guidelines for the clinical management of patients for the prevention of atherosclerotic cardiovascular disease (ASCVD) risk [4].

The use of proprotein convertase subtilisin/kexin type 9 serine protease (PCSK9) inhibitors [5,6] has made the measurement of low LDL-C critical for the secondary prevention of ASCVD. Because of its expense, PCSK9-inhibitors are typically reserved for high-risk ASCVD patients who do not achieve LDL-C levels below at least 70 mg/dL on more conventional therapy [4,7,8]. Most clinical laboratories still use the Friedewald equation (F-LDL-C) to calculate LDL-C based on the results of the standard lipid panel (total cholesterol (TC), triglycerides (TG) and high-density lipoprotein cholesterol (HDL-C)) [9,10]. Typically, F-LDL-C closely matches LDL-C as determined by the β-quantification reference method (BQ), a laborious combined precipitation-ultracentrifugation procedure [11,12]. The F-LDL-C equation is known, however, to underperform for hypertriglyceridemic (HTG) samples (TG > 400 mg/dL), because its TG/5 term overestimates cholesterol on very-low-density lipoproteins (VLDL), leading to an underestimation of LDL-C [13,14,15,16]. 

Because of the known limitations of the F-LDL-C equation, the current US-Multi-society Cholesterol Guideline on the Management of Blood Cholesterol [4] recommends that either a direct LDL-C test or an alternative LDL-C equation be used when LDL-C is low (<70 mg/dL). Although direct LDL-C tests are now fully automated and widely available, they can differ from the BQ reference method for various types of dyslipidemia, including HTG [10]. Because of this issue, which is related to their differential reactivity to different lipoprotein subfractions, and the extra costs for performing direct LDL-C testing, most clinical laboratories in the US still calculate LDL-C, according to recent College of American Pathology proficiency test surveys. In 2018, the US-Multi-society Cholesterol Guideline [4] recommended “enhanced equations” such as the Martin equation (M-LDL-C) [15,17] rather than F-LDL-C for estimating LDL-C when concentrations are low. The M-LDL-C equation, designed to match LDL-C measured by the vertical auto profile (VAP) ultracentrifugation method [18,19] is identical to the F-LDL-C equation except for its TG denominator, which varies depending upon the plasma levels of TG and non-high-density lipoprotein cholesterol (nonHDL-C) [15,17,20]. Recently, a modified Martin equation (extended Martin equation; eM-LDL-C) was described with a different set of TG denominators for TG between 400 and 800 mg/dL [21]. 

The accurate measurement of LDL-C at the high end is also clinically relevant, particularly for primary prevention. According to the US-Multi-society Cholesterol Guideline [4], patients with LDL-C > 190 mg/dL do not need to undergo any further ASCVD risk assessment and should be treated with a statin. For patients with HTG, it is recommended that a nonHDL-C cut-point of 220 mg/dL, which can be accurately calculated by a simple calculation, be used instead for deciding statin therapy, because of potential inaccuracies in LDL-C estimation [4]. Some have also advocated more widespread use of nonHDL-C as an ASCVD biomarker, but current guidelines still focus most of their recommendations based on LDL-C values. 

In 2020, we described a bivariate quadratic equation, called the Sampson equation (S-LDL-C) [16], designed to match LDL-C measured by the BQ reference method [11,12]. Overall, it was more accurate than the other LDL-C equations when compared to BQ, particularly for high TG samples up to 800 mg/dL [16]. In this study, we compare the S-LDL-C equation to the two different versions of the Martin equation and the F-LDL-C equation against the BQ reference method for both low and high LDL-C values. We also describe a new method for assessing the clinical impact of inaccuracies in LDL-C estimation methods, using error-grid analysis [22].

## 2. Methods

Deidentified LDL-C and other lipid test results were obtained from the clinical laboratory at Mayo Clinic on patients (N = 40,346) for whom BQ testing was performed as previously described [23,24]. Samples with detectable Lipoprotein-X by agarose gel electrophoresis (N = 141), with TG > 2000 mg/dL (N = 172), with TC > 1000 mg/dL (N = 6), or with Type III hyperlipidemia (TG between 150 and 1000 mg/dL with measured VLDL-C/TG > 0.3, N = 71) were excluded from analysis. The mean and range of lipid values and patient demographic information for the final dataset are shown in Appendix A.

LDL-C was calculated by the F-LDL-C [9], M-LDL-C [17], eM-LDL-C [21] and S-LDL-C [16] equations (Appendix A) by an Excel spreadsheet, which can be downloaded at the following website: https://figshare.com/articles/software/Sampson_LDLC_and_VLDLC_calculator/21346893. The overall concordance with the BQ method for classification by the different equations into LDL-C intervals was determined by either calculating the balanced-accuracy (BA) index (Sensitivity + Specificity/2) or the normalized Matthews correlation coefficient (nMCC) index, as previously described [25]. Comparisons among LDL-C equations for the number of potentially clinically relevant errors were done by pairwise Chi-Square analysis and by calculating their kappa scores. Research under this study was not considered human subject research and was exempted from IRB review.

## 3. Results

We first compared the various LDL-C equations against the BQ reference method (BQ-LDL-C) by regression analysis on a large number of patients with a wide range of LDL-C values (Figure 1). Based on their mean absolute difference (MAD) and other metrics of test accuracy (slope, intercept, correlation coefficient (R^2^) and root mean square error (RMSE)), the S-LDL-C equation (Figure 1D) showed greater accuracy than the F-LDL-C (Figure 1A), M-LDL-C (Figure 1B), or eM-LDL-C equations (Figure 1C). The eM-LDL-C equation was only slightly more accurate than the original M-LDL-C equation in the whole dataset, but when results with TG 400–800 mg/dL were separately analyzed (Appendix A), there was greater improvement over the original Martin equation (M-LDL-C MAD = 27.1, eM-LDL-C MAD = 24.5). Nonsensical negative LDL-C values for high TG samples occurred mostly with the F-LDL-C equation (Figure 1A). An analysis of all equations by their residual errors as a function of the main independent variables (TG, nonHDL-C and HDL-C) as well as apoB and age also indicated S-LDL-C had the smallest residual errors, followed by eM-LDL-C, M-LDL-C and F-LDL-C (Appendix A). 

A plot of MAD for the four equations against the BQ reference method for different intervals of TG and nonHDL-C is shown in Figure 2. In HTG samples, greater accuracy was observed for S-LDL-C compared to the other equations (Figure 2A). At a TG interval centered at 400 mg/dL, the F-LDL-C equation had a MAD score of approximately 20 mg/dL, which we used as a benchmark because the Friedewald equation is not recommended for samples with TG exceeding this value because of inaccuracy. The S-LDL-C equation crosses this threshold at a TG level between 800 and 1000 mg/dL, whereas the original Martin equation exceeds this threshold between a TG level of 390 and 410 mg/dL. The extended Martin equation exceeded this threshold at a slightly higher TG level somewhere between 410 and 500 mg/dL. When the different equations were examined for different intervals of nonHDL-C, the S-LDL-C equation again appeared to be the most accurate, particularly for high nonHDL-C samples. The two Martin equations were the least accurate (Figure 2B). Using the same 20 mg/dL LDL-C error threshold used for the different TG intervals, it appears that the S-LDL-C equation can be used for nonHDL-C values up to at least 350 mg/dL.

To assess the accuracy of the equations for estimating low LDL-C, regression analysis was performed on low LDL-C samples (<100 mg/dL) for those with TG 400–800 mg/dL and <400 mg/dL. By all the different accuracy metrics, S-LDL-C had the best overall performance for HTG samples, followed by eM-LDL-C and M-LDL-C and finally F-LDL-C (Figure 3). Both the M-LDL-C and eM-LDL-C equations exhibited a fixed positive bias, as can be seen by their relatively large positive intercepts and how their regression lines were above and parallel to the line of identity. In contrast, the F-LDL-C equation showed a negative bias, particularly for HTG patients with low LDL-C values, which sometimes resulted in negative LDL-C values. 

When samples with low LDL-C and TG < 400 mg/dL were analyzed (Figure 4), the LDL-C equations were more similar in their performance, but they maintained the same rank order in their accuracy. Note that only results of the M-LDL-C equation are shown, because it yields identical results to the eM-LDL-C equation for TG < 400 mg/dL. Further subdivision of TG to <175 mg/dL versus 175–400 mg/dL revealed a slight negative bias for F-LDL-C for samples with TG 175–400 mg/dL. In contrast, the M-LDL-C equation showed a slight positive bias for those same samples with modest TG elevations.

To evaluate the different LDL-C equations for high LDL-C samples, we performed regression analysis against BQ-LDL-C for LDL-C between 160 and 220 mg/dL to bracket the 190 mg/dL high cut-point recommended for primary prevention screening (Figure 5). Based on this analysis, all the equations showed better performance at the high LDL-C cut-point, but the S-LDL-C equation was again slightly better by most of the accuracy metrics followed by the F-LDL-C and then the two Martin equations. When samples with TG 400–800 mg/dL were analyzed separately, it was observed that the M-LDL-C and eMLDL-C equations had a positive bias of at least 20 mg/dL, as can be observed by their positive regression line across the whole LDL-C 160–220 mg/dL test interval. Improved accuracy of the S-LDL-C equation for high LDL-C samples was also demonstrated by analysis of a larger sample set with LDL-C ranging between 100 and 700 mg/dL (Appendix A).

Next, for patients with TG 400–800 mg/dL, we used error grid analysis [22] to compare the analytic errors of the different LDL-C equations for their potential to change clinical management decisions. As shown in Figure 6A, differences between estimated LDL-C and BQ-LDL-C that were greater than the 12% proportional total allowable error goal for LDL-C [10] but not expected to change clinical management (no change in classification at the low (70 mg/dL) and high (190 mg/dL), were categorized as pure analytical errors. Errors that resulted in the incorrect classification of a patient at either the low or high LDL-C cut-point were classified as clinically relevant errors regardless of the magnitude of the difference between the estimated and BQ LDL-C values. For TG 400–800 mg/dL, only approximately half of the S-LDL-C results were analytically correct (within the 12% total allowable error goal), but this was much better than the other equations (Figure 6F). Likewise, the S-LDL-C equation had the least analytically incorrect results. Its errors were also more balanced than the other equations. F-LDL-C more often underestimated true LDL-C, whereas M-LDL-C and eM-LDL-C more frequently overestimated LDL-C. In terms of clinically relevant errors (Figure 6H), a total of 13.5% of the S-LDL-C results would be predicted to potentially change the management of patients, which was statistically less than for F-LDL-C (23.0%), M-LDL-C (20.5%) and eM-LDL-C (20.0%) (Appendix A). The clinically relevant errors for F-LDL-C tended to underestimate LDL-C at the low LDL-C cut-point, whereas M-LDL-C and eM-LDL-C more often overestimated LDL-C at both the low and high LDL-C cut-points.

Similar error-grid analysis performed for patients with TG < 400 mg/dL indicated smaller differences between the equations (Figure 7). Much higher percentages of results were analytically correct (Figure 7D) and fewer were analytically incorrect with limited clinical impact (Figure 7E). In terms of clinically relevant errors at the high LDL-C cut-point, all 4 equations were similar in performance (Figure 7F). A greater percentage of clinically relevant errors was observed at the low LDL-C cut-point, but again all equations were similar in performance except for F-LDL-C, which statistically had the greatest frequency of errors due to an underestimation of LDL-C (Appendix A).

Finally, we calculated in Table 1 the concordance of the four equations for classification of patients into a variety of previously recommended LDL-C treatment intervals [4,26]. For each LDL-C interval, spanning low to high LDL-C values, we calculated true positive, true negative, false positive and false negative test results when compared against the BQ reference method. Using these four possible test outcomes, we also calculated the positive and negative predictive value for each equation, as well as their sensitivity and specificity for correctly classifying patients into their true LDL-C interval as determined by the BQ reference method. For an overall index, we calculated the BA and nMCC index scores. For TG < 400 mg/dL, S-LDL-C had the best BA index for all LDL-C intervals. Similarly, the S-LDL-C equation had the best nMCC index for all LDL-C intervals except for 40–69 mg/dL, which was slightly better for the M-LDL-C equation. In general, all four equations showed relatively good performance for low TG samples and classification differences between the different equations were relatively small. In contrast, for samples with TG 400–800 mg/dL, the S-LDL-C equation was more concordant with the BQ reference method for all of the LDL-C intervals tested based on both the BA and nMCC indices. 

## 4. Discussion

Because of the clinical need to accurately measure both high and low LDL-C, it is a challenge to develop a single equation that shows adequate accuracy on both ends of the LDL-C reference range. In fact, the Friedewald equation was first developed over 50 years ago when the main clinical concern was only high LDL-C [9]. Only recently with new effective therapies such as PCSK9-inhibitors have we been able to routinely lower LDL-C below 70 mg/dL or even lower, which has now become a goal for secondary prevention [4].

Although the M-LDL-C equation was first reported in 2013 [20], recent College of American Pathologist Clinical Chemistry Surveys indicate that the majority of clinical laboratories still use the F-LDL-C equation. In 2018, the Multi-society Cholesterol Guidelines [4] specifically recommended that the M-LDL-C equation [15,20] be used for low LDL-C samples but did not comment on the use of F-LDL-C equation for other types of samples. Results from this study and now many other studies [10,27,28,29] have clearly shown that the F-LDL-C equation does not offer any advantages over more recently developed equations for calculating LDL-C. It may take a more explicit recommendation from future US guidelines discouraging the use of the F-LDL-C equation, at least for samples with more than modest elevations in TG, before more clinical laboratories will switch their LDL-C calculation method. An expert panel from the Canadian Society of Clinical Chemists did recently recommend that the F-LDL-C equation be replaced with the S-LDL-C equation for routine use [30].

There are two potential barriers that may have slowed the replacement of the F-LDL-C equation by the M-LDL-C or eM-LDL-C equations, which are not an issue with the S-LDL-C equation. First, the S-LDL-C equation can be directly and easily implemented by most clinical laboratory information systems, because they are all typically designed for user entry of novel equations. In contrast, custom software changes for some laboratory information systems may be needed to implement the 180-cell look-up tables of TG denominators that are required for the M-LDL-C and eM-LDL-C equations. Secondly, the S-LDL-C equation is in the public domain and is free to use without any fees or other type of restrictions. The method for calculating LDL-C by the M-LDL-C equation has been patented and is licensed to Quest Diagnostics. 

In terms of accuracy, the Martin and Sampson equations appear to yield similarly accurate results for most samples, but S-LDL-C appears to have a clear advantage for HTG samples even when compared to the new eM-LDL-C equation. As we show by error-grid analysis, the S-LDL-C equation also results in fewer clinically relevant errors compared to the other equations, particularly for HTG samples. The improved accuracy of the S-LDL-C equation over the M-LDL-C and eM-LDL-C equations may be a consequence of the method used to measured LDL-C when developing the Martin equations. The S-LDL-C equation was trained against the BQ reference method, whereas the original and new enhanced Martin equations were based on the VAP method [19]. Both VAP and BQ utilize ultracentrifugation to separate lipoproteins; however, the VAP method has been reported to under-recover TG-rich lipoproteins (VLDL and intermediate-density lipoproteins (IDL)) compared to the BQ reference method and was the reason that this method was not recommended for HTG samples when first developed [19,31,32]. Because LDL-C is calculated by the M-LDL-C equation by subtracting HDL-C and VLDL-C from TC, any under-recovery of VLDL-C by the VAP method would be expected to lead to the observed positive bias in LDL-C for high TG samples by both Martin equations. 

When possible, it is, of course, always best to evaluate a method by comparing it to its reference method, which ideally all routine test methods in the field are traced against. Furthermore, in the case of lipids, almost all initial clinical trials of lipid-lowering agents utilized the BQ reference method for establishing the link between lipid lowering and clinical outcomes. Many recent studies [33], however, comparing the different LDL-C equations, have used a direct LDL-C assay to assess accuracy and have sometimes come to different conclusions about the relative accuracy of different equations. Although direct LDL-C assays are sometimes used for HTG samples because of their improved accuracy, they can nevertheless still have significant positive or negative biases [34], which can lead to differences in the interpretation of the accuracy of the various LDL-C equations. Given that the various LDL-C equations yield similar results for most samples, it is also important to evaluate a relatively large number of samples, as was done in this study. It is particularly important to assess patients with HTG and other types of dyslipidemia to fully evaluate the accuracy of the different LDL-C equations [34]. In terms of the difference between the M-LDL-C and eM-LDL-C equations, we found only a relatively modest improvement in the accuracy of the eM-LDL-C equation for HTG samples when both methods were compared against the BQ reference method. Again, this highlights the importance of evaluating any new method for estimating LDL-C against the BQ reference method, which was not done when initially developing the eM-LDL-C equation [21].

Another important issue is the best way to assess the accuracy of classifying patients into different LDL-C treatment intervals. The M-LDL-C equation was previously assessed for its classification concordance with the BQ reference method by its ratio of true positives over true positives plus false positives [15], which is its positive predictive value. By itself, positive predictive value is known, however, to be a potentially misleading index of test classification accuracy. It does not take into account false negative test results and is, therefore, unaffected by prevalence [35]. If one does use positive predictive value for this purpose, it is then important to also consider negative predictive value in conjunction with positive predictive value. Alternatively, sensitivity in conjunction with specificity can also be used to assess test concordance with a reference method and is the more conventional way for evaluating diagnostic test performance [36]. There are, however, several different indices of overall test accuracy, each with their own advantages and disadvantages [37]. We used both the BA index, which weighs sensitivity and specificity equally, and the nMCC index, which can weigh sensitivity and specificity differently to account for any imbalance in the number of true positive and true negatives [25]. In our case, both metrics yielded a similar interpretation, indicating an advantage of the S-LDL-C equation over the other equations, particularly for HTG samples. 

Another way to assess the accuracy of LDL-C equations is by error-grid analysis [22], which was previously used for evaluating glucose monitors, but we modified it for LDL-C equation assessment. It is a hybrid approach that allows one to separately consider purely analytical errors versus clinically relevant errors. Based on this analysis, the S-LDL-C equation resulted in fewer clinically relevant errors than the other equations for HTG at the low (LDL-C < 70 mg/dL) and high (LDL-C > 190 mg/dL) cut-points. For TG < 400 mg/dL, S-LDL-C and M-LDL-C had similar frequency of clinically relevant errors and F-LDL-C had the most. These results are consistent with a recent report based on the Canadian Health Measure Survey showing that the replacement of F-LDL-C with the S-LDL-C equation is justified based on the number of patients for whom it would affect either the initial decision to treat with a statin or statin dose [38]. 

In summary, the F-LDL-C equation does not appear to have any advantages over the other LDL-C equations and should be replaced with one of the newer alternative LDL-C equations. The use of more accurate alternative LDL-C equations would likely most benefit those patients who may need to receive a second lipid-lowering agent in order to reduce any remaining high residual risk. For most samples, the alternative LDL-C equations showed similar performance, but S-LDL-C is the most accurate on samples with more than moderate levels of HTG and has several practical advantages in terms of ease of implementation. A limitation of our study is that we only have information on the age and sex of our patients, so it will be important to assess the different LDL-C equations in different ethnic populations and in patients with specific medical disorders to determine if our results are generalizable. Additionally, even though the BQ method is the reference method, it is important to note that cholesterol in the fraction it classifies as LDL also includes cholesterol on Lp(a) and some remnant lipoproteins too. In the future, it would, therefore, be important to directly assess the different LDL-C equations, which may be affected differently by cholesterol on these other lipoproteins, for their impact in the clinical management of patients and for their ability to predict future ASCVD events.

## Figures and Tables

**Figure 1 biomedicines-10-03156-f001:**
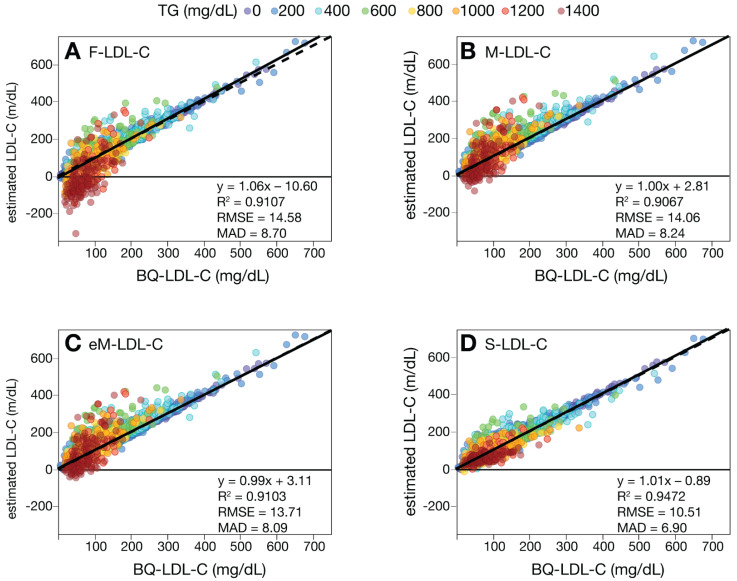
Comparison of estimated LDL-C versus BQ-LDL-C. LDL-C was calculated in patients (N = 39,956) with a wide range of LDL-C values by F-LDL-C, (Panel **A**), M-LDL-C (Panel **B**), eM-LDL-C (Panel **C**) and S-LDL-C (Panel **D**) equations and plotted against LDL-C as measured by BQ reference method (BQ-LDL-C). Solid lines are linear fits for the indicated regression equations. Dotted lines are lines of identity. Results are color coded by TG level with the value in the legend (mg/dL) indicating the start of each interval.

**Figure 2 biomedicines-10-03156-f002:**
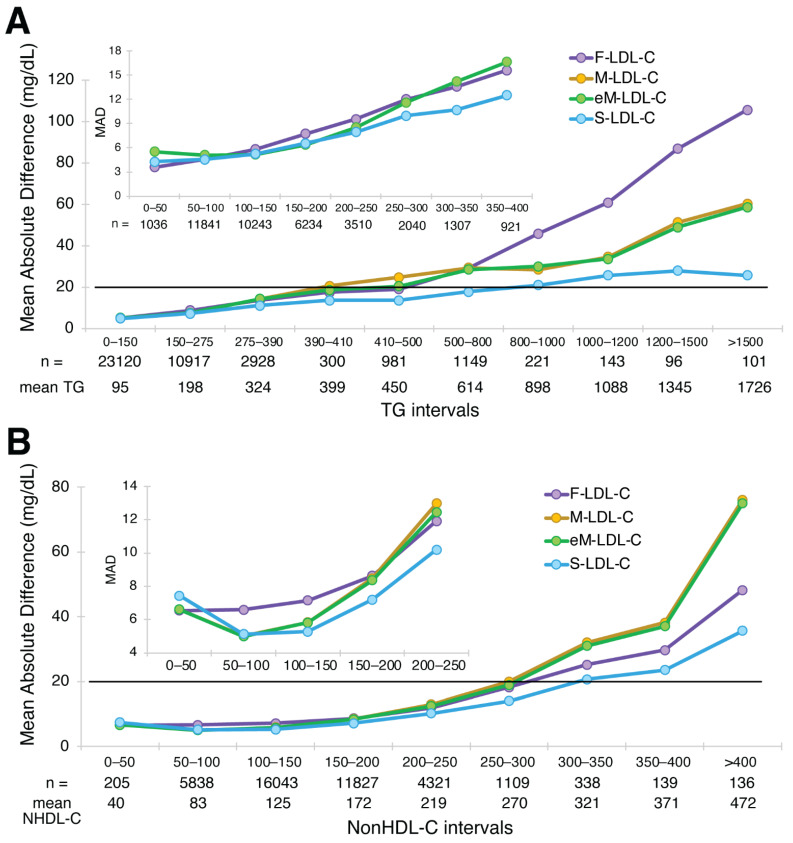
Mean Absolute Difference of estimated LDL-C versus BQ-LDL-C. Mean absolute difference (MAD) score for LDL-C from patients (N = 39,956) with a wide range of LDL-C values is shown for the F-LDL-C (purple line), the M-LDL-C (orange line), eM-LDL-C (green line) and S-LDL-C (light blue line) equations for the indicated TG intervals (Panel **A**) and nonHDL-C intervals (Panel **B**). The inset shows a close-up for low TG and low nonHDL-C samples. The number of samples within the interval is indicated, as well as the mean value for the interval. Solid black line is the level of the MAD for Friedewald at 400 mg/dL TG (20 mg/dL), which was used as a limit for acceptable accuracy for the other equations.

**Figure 3 biomedicines-10-03156-f003:**
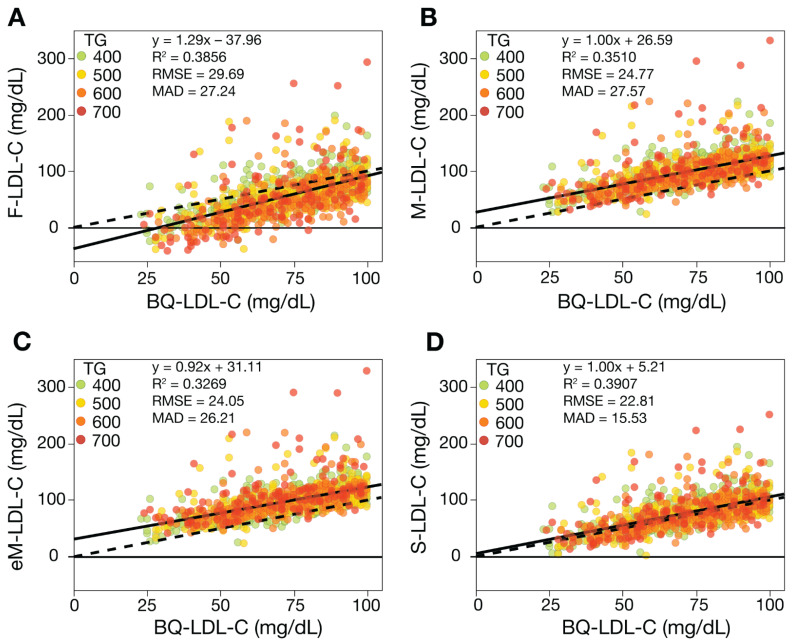
Comparison of estimated LDL-C versus BQ-LDL-C for HTG samples with low LDL-C. LDL-C was calculated from patients (N = 1115) with LDL-C < 100 mg/dL and TG 400–800 mg/dL values by F-LDL-C (Panel **A**), M-LDL-C (Panel **B**), eM-LDL-C, (Panel **C**) and S-LDL-C (Panel **D**) equations and plotted against LDL-C as measured by the BQ reference method (BQ-LDL-C). Solid lines are the linear fits for the indicated regression equations. Dotted lines are lines of identity. Results are color coded by TG level with the value in the legend (mg/dL) indicating the start of each interval.

**Figure 4 biomedicines-10-03156-f004:**
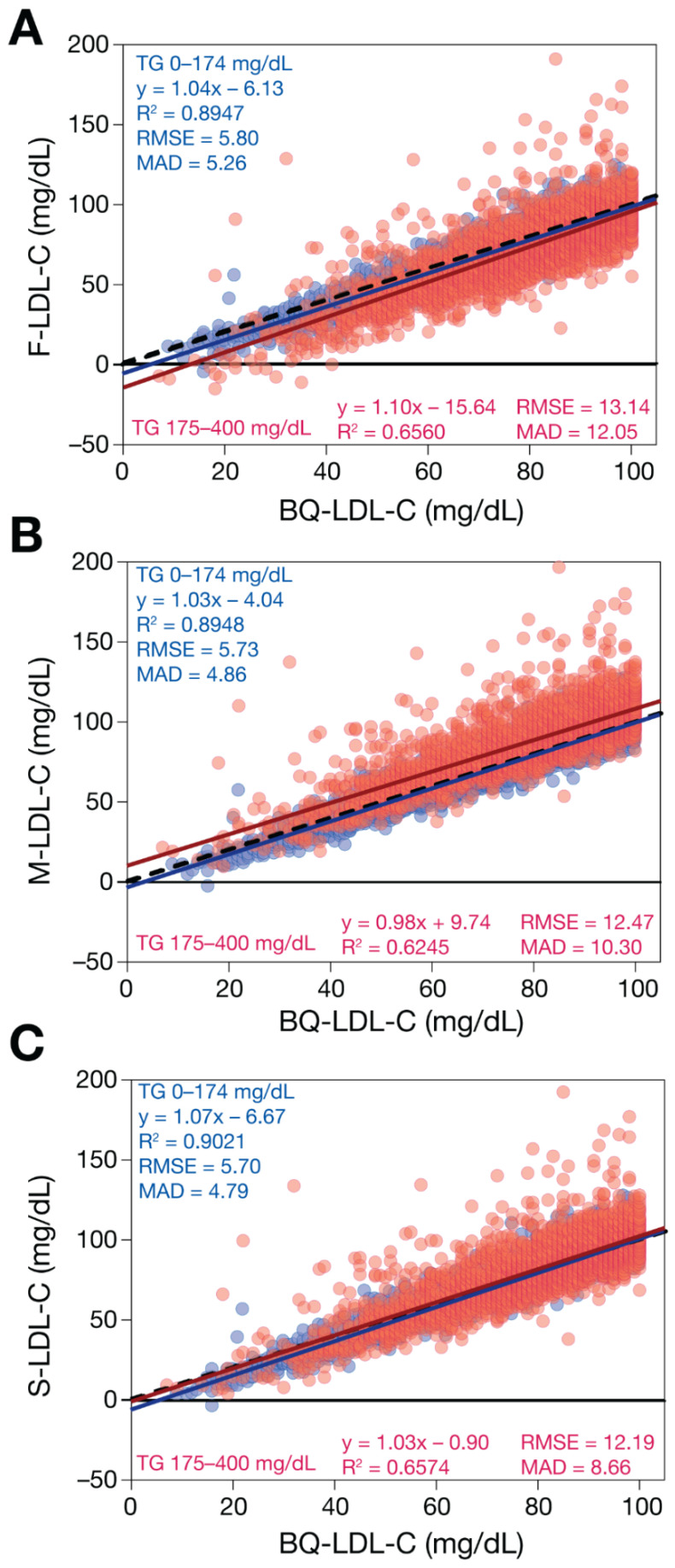
Comparison of estimated LDL-C versus BQ-LDL-C for low TG samples with low LDL-C. LDL-C was calculated for patients (N = 13,415) with LDL-C < 100 mg/dL and TG < 400 mg/dL values by F-LDL-C (Panel **A**), M-LDL-C (Panel **B**), and S-LDL-C (Panel **C**) equations and plotted against LDL-C as measured by the BQ reference method (BQ-LDL-C). Solid lines are the linear fits for the indicated regression equations. Dotted lines are lines of identity. Results are color coded by TG level with TG < 175 mg/dL indicated in blue and samples with TG between 175 and 400 mg/dL in red.

**Figure 5 biomedicines-10-03156-f005:**
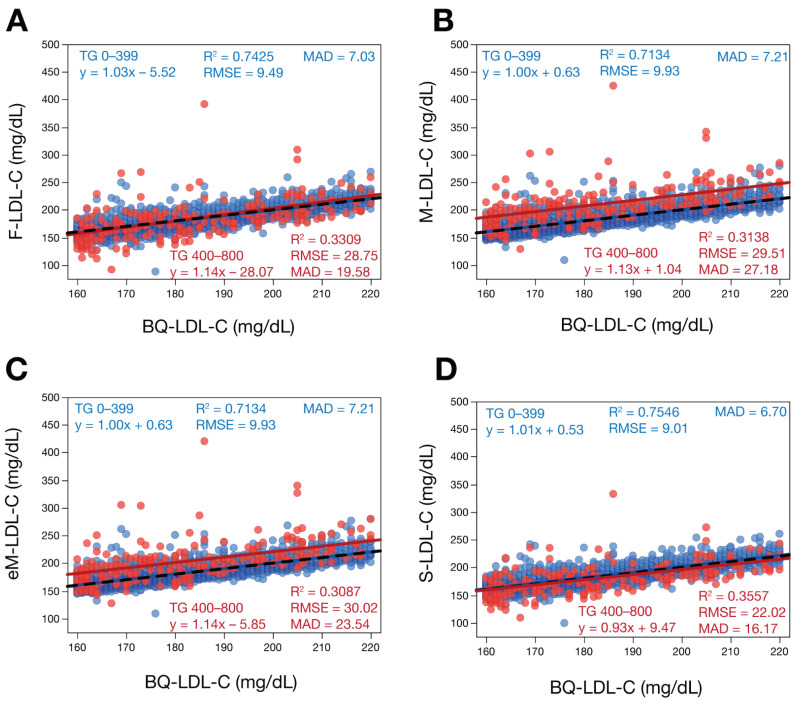
Comparison of estimated LDL-C versus BQ-LDL-C for samples with high LDL-C. LDL-C was calculated for patients (N = 5060) with LDL-C between 160 and 220 mg/dL by F-LDL-C (Panel **A**), M-LDL-C (Panel **B**), eM-LDL-C (Panel **C**) and S-LDL-C (Panel **D**) equations and plotted against LDL-C as measured by BQ reference method (BQ-LDL-C). Solid red lines are the linear fits for the indicated regression equations for samples with TG > 400 mg/dL. Dotted lines are lines of identity. Results are color coded by TG level with TG < 400 mg/dL indicated in blue and TG 400–800 mg/dL in red.

**Figure 6 biomedicines-10-03156-f006:**
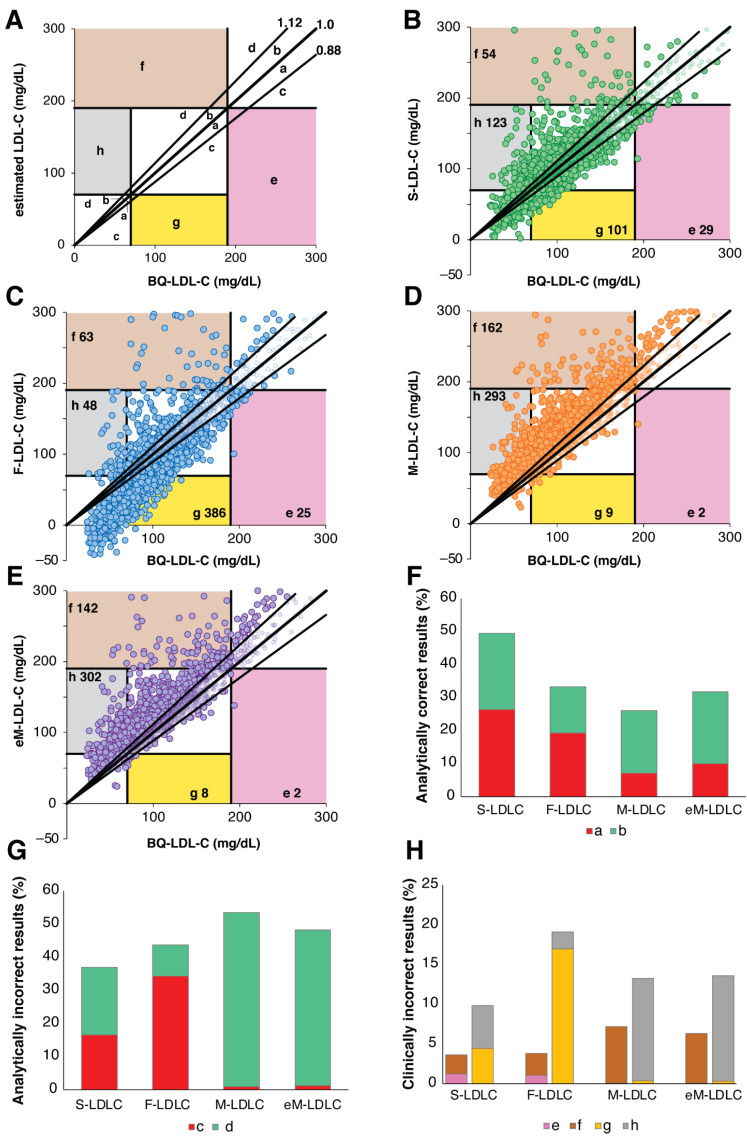
Error Grid Analysis for high TG samples. Definition of type of errors are shown in (Panel **A**). a: Within 12% proportional error and below regression line, b: Within 12% proportional error and above regression line, c: Greater than 12% proportional error but no impact on patient management and below regression line, d: Greater than 12% proportional error but no impact in patient management and above regression line, e: Underestimation of LDL-C at high LDL-C cut-point leading to error in patient management, f: Overestimation of LDL-C at high LDL-C cut-point leading to error in patient management, g: Underestimation of LDL-C at low LDL-C cut-point leading to error in patient management, h: Overestimation of LDL-C at low LDL-C cut-point leading to error in patient management. Numbers in colored zones (e, f, h and g) indicate total number of clinically relevant misclassifications. Error grid analysis was performed on patients (N = 2274) with TG 400–800 mg/dL and BQ-LDL-C ≤ 300 mg/dL for LDL-C calculated by the S-LDL-C (Panel **B**), F-LDL-C (Panel **C**), M-LDL-C (Panel **D**), and eM-LDL-C (Panel **E**) equations. Percent of analytically correct results within 12% proportional error (Panel **F**, Zones a + b) and incorrect analytical results (Panel **G**, Zones c + d) are shown. Clinically relevant errors affecting classification at high (Zones e + f) and low (Zones g + h) LDL-C cut-points are shown in (Panel **H**).

**Figure 7 biomedicines-10-03156-f007:**
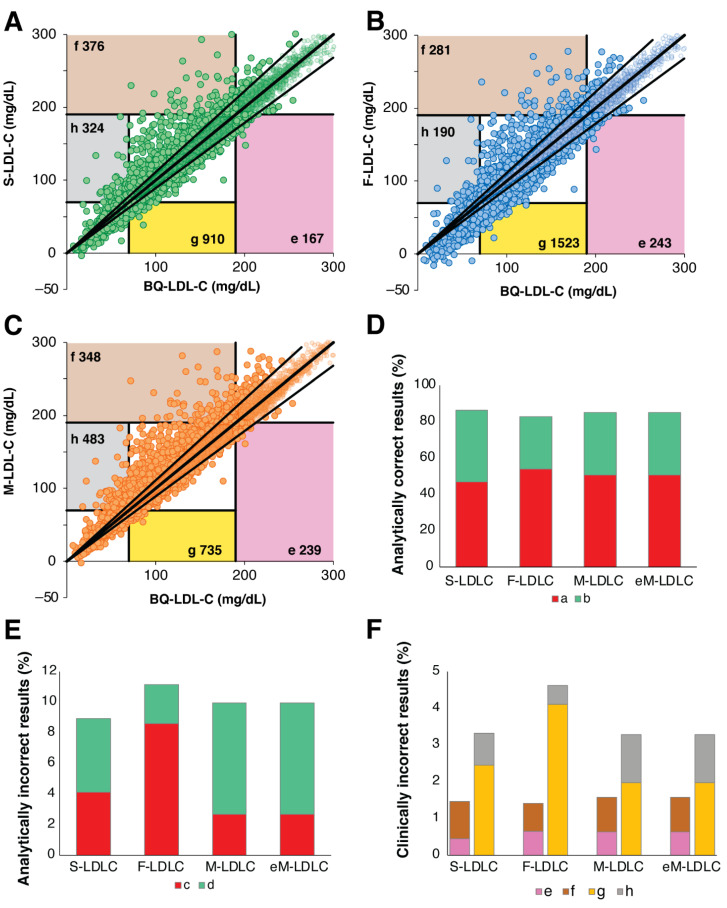
Error Grid Analysis for low TG samples. Definition of type of errors are the same as shown in Figure 6A. Error grid analysis was performed for patients (N = 37,088) with TG < 400 mg/dL and BQ-LDL-C ≤ 300 mg/dL for LDL-C calculated by the S-LDL-C (Panel **A**), F-LDL-C (Panel **B**), M-LDL-C (Panel **C**) equations. Percent of analytically correct results within 12% proportional error (Panel **D**, Zones a + b) and incorrect analytical results (Panel **E**, Zones c + d) are shown. Clinically relevant errors affecting classification at low (Zones e + f) and high (Zones g + h) LDL-C cut-points are shown in (Panel **F**).

**Table 1 biomedicines-10-03156-t001:** Concordance of LDL-C equations with BQ for classification into LDL-C intervals.

TG 0–400 mg/dL										
	TP	TN	FP	FN	ppv	npv	Sensitivity	Specificity	BA	nMCC
*BQ*-LDL-C 40–69 mg/dL										
*F*-LDL-C	2719	32,342	1495	594	64.5	98.2	82.1	95.6	88.8	0.849
*M*-LDL-C	2675	33,077	760	638	77.9	98.1	80.7	97.8	89.2	0.886
*S*-LDL-C	2770	32,912	925	543	75.0	98.4	83.6	97.3	90.4	0.885
***BQ*-LDL-C 70–99 mg/dL**										
*F*-LDL-C	7320	25,613	2173	2044	77.1	92.6	78.2	92.2	85.2	0.850
*M*-LDL-C	7471	26,108	1678	1893	81.7	93.2	79.8	94.0	86.9	0.872
*S*-LDL-C	7544	26,248	1538	1820	83.1	93.5	80.6	94.5	87.5	0.879
***BQ*-LDL-C 100–129 mg/dL**										
*F*-LDL-C	8405	24,196	1807	2742	82.3	89.8	75.4	93.1	84.2	0.851
*M*-LDL-C	8643	23,944	2059	2504	80.8	90.5	77.5	92.1	84.8	0.852
*S*-LDL-C	8765	24,342	1661	2382	84.1	91.1	78.6	93.6	86.1	0.868
***BQ*-LDL-C 130–159 mg/dL**										
*F*-LDL-C	5636	28,282	1447	1785	79.6	94.1	75.9	95.1	85.5	0.862
*M*-LDL-C	5747	27,857	1872	1674	75.4	94.3	77.4	93.7	85.6	0.852
*S*-LDL-C	5886	28,113	1616	1535	78.5	94.8	79.3	94.6	86.9	0.868
***BQ*-LDL-C 160–189 mg/dL**										
*F*-LDL-C	2620	32,848	759	923	77.5	97.3	73.9	97.7	85.8	0.866
*M*-LDL-C	2680	32,592	1015	863	72.5	97.4	75.6	97.0	86.3	0.856
*S*-LDL-C	2785	32,665	942	758	74.7	97.7	78.6	97.2	87.9	0.871
**TG 401–800 mg/dL**										
	**TP**	**TN**	**FP**	**FN**	**ppv**	**npv**	**sensitivity**	**specificity**	**BA**	**nMCC**
***BQ*-LDL-C 40–69 mg/dL**										
*F*-LDL-C	111	1543	312	283	26.2	84.5	28.2	83.2	55.7	0.555
*M*-LDL-C	110	1814	41	284	72.8	86.5	27.9	97.8	62.9	0.695
*eM*-LDL-C	103	1815	40	291	72.0	86.2	26.1	97.8	62.0	0.687
*S*-LDL-C	218	1736	119	176	64.7	90.8	55.3	93.6	74.5	0.760
***BQ*-LDL-C 70–99 mg/dL**										
*F*-LDL-C	224	1354	249	422	47.4	76.2	34.7	84.5	59.6	0.606
*M*-LDL-C	205	1371	232	441	46.9	75.7	31.7	85.5	58.6	0.599
*eM*-LDL-C	236	1357	246	410	49.0	76.8	36.5	84.7	60.6	0.617
*S*-LDL-C	374	1384	219	272	63.1	83.6	57.9	86.3	72.1	0.727
***BQ*-LDL-C 100–129 mg/dL**										
*F*-LDL-C	241	1497	191	320	55.8	82.4	43.0	88.7	65.8	0.674
*M*-LDL-C	197	1283	405	364	32.7	77.9	35.1	76.0	55.6	0.554
*eM*-LDL-C	250	1282	406	311	38.1	80.5	44.6	75.9	60.3	0.598
*S*-LDL-C	306	1452	236	255	56.5	85.1	54.5	86.0	70.3	0.705
***BQ*-LDL-C 130–159 mg/dL**										
*F*-LDL-C	121	1819	122	187	49.8	90.7	39.3	93.7	66.5	0.683
*M*-LDL-C	146	1567	374	162	28.1	90.6	47.4	80.7	64.1	0.615
*eM*-LDL-C	181	1616	325	127	35.8	92.7	58.8	83.3	71.0	0.673
*S*-LDL-C	175	1752	189	133	48.1	92.9	56.8	90.3	73.5	0.720
***BQ*-LDL-C 160–189 mg/dL**										
*F*-LDL-C	66	2027	81	75	44.9	96.4	46.8	96.2	71.5	0.711
*M*-LDL-C	55	1920	188	86	22.6	95.7	39.0	91.1	65.0	0.617
*eM*-LDL-C	58	1977	131	83	30.7	96.0	41.1	93.8	67.5	0.653
*S*-LDL-C	78	2015	93	63	45.6	97.0	55.3	95.6	75.5	0.733

## Data Availability

All data available upon request.

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
