# Peer review of "Accuracy and Clinical Impact of Estimating Low-Density Lipoprotein-Cholesterol at High and Low Levels by Different Equations"

_biomedicines, 2022, doi:10.3390/biomedicines10123156_

Round 1

Reviewer 1 Report

This is an excellent comparison of different approaches to calculate LDL-cholesterol usung different formulas.   

1. There is a ß missing in line 18 page 1 and line 49 page 2 in the PDF that I downloaded.

2. Before considering the reference method or gold standard for LDL quantification, it would be useful to simply define LDL. According to the definition of the Lipid Research Clinic's protocol, LDL refers to lipoporteins of density (d)=1.006-1.063 g/ml. In this approach, LDL comprises IDL (d=1.006-1.019 g/ml).  3. The ß-quantification method is the 'reference method', which the authors consider to be the gold standard. Plasma or serum is centrifuged overnight at d=1.006 g/ml to spin up VLDL. The apolipoprotein B-containing lipoproteins in the bottom fraction are precipitated using heparin and manganese chloride, leaving HDL in solution. LDL cholesterol is calculated by subtracting VLDL-cholesterol and HDL-cholesterol from total cholesterol. Therefore, in the reference method, LDL contains  IDL, LDL in the strict sense, and Lp(a). The reference method is distinct from the 'real' gold standard, which is sequential ultracentrifugation (?) and which is cumbersome to be applied on a large number of samples. My point is that ‘LDL’ has different meanings. Clearly, in the Friedewald formula, IDL is also calculated as LDL. 4. IDL-cholesterol is generally quite low compared to LDL-cholesterol in the strict sense. Direct methods (homogeneous methods) show only partial cross-reactivity with IDL. Is there any place for direct methods in the clinical laboratory. These assays exist for more than 20 years but they simply increase confusion. 5. Would it not be more practical to quantify non-HDL cholesterol? Many will argue that remnant cholesterol should be quantified separately. If so, which method should be used? Calculation or method of direct remant cholesterol quantification?

Author Response

This is an excellent comparison of different approaches to calculate LDL-cholesterol using different formulas.

Thank you for your overall comments and careful review of the paper.

  1. There is a ß missing in line 18 page 1 and line 49 page 2 in the PDF that I downloaded.

We think this is a mistake in formatting related to making the PDF and will work with the copy editor to make these corrections.

  1. Before considering the reference method or gold standard for LDL quantification, it would be useful to simply define LDL. According to the definition of the Lipid Research Clinic's protocol, LDL refers to lipoporteins of density (d)=1.006-1.063 g/ml. In this approach, LDL comprises IDL (d=1.006-1.019 g/ml). 

Thank you for the helpful suggestion.  We have now defined LDL in the introduction.  See the below changes to the revised manuscript:

Lines 35-36: Cholesterol in low-density lipoproteins (LDL) (density range: 1.006-1063 g/mL), is causally related to the development of atherosclerosis [1].

  1. The ß-quantification method is the 'reference method', which the authors consider to be the gold standard. Plasma or serum is centrifuged overnight at d=1.006 g/ml to spin up VLDL. The apolipoprotein B-containing lipoproteins in the bottom fraction are precipitated using heparin and manganese chloride, leaving HDL in solution. LDL cholesterol is calculated by subtracting VLDL-cholesterol and HDL-cholesterol from total cholesterol. Therefore, in the reference method, LDL contains  IDL, LDL in the strict sense, and Lp(a). The reference method is distinct from the 'real' gold standard, which is sequential ultracentrifugation (?) and which is cumbersome to be applied on a large number of samples. My point is that ‘LDL’ has different meanings. Clearly, in the Friedewald formula, IDL is also calculated as LDL.

Yes, the reviewer is correct about the limitations of the ß-quantification method.  Nevertheless, it is the reference method used by the CDC and other organizations throughout the world to which routine methods are traced against and is the method that the diagnostic companies use in the calibration of their routine lipid assays.  We have now acknowledged some of the limitations of the ß-quantification method in the revised discussion.  See below.

Lines 377-382: Also, even though the BQ method is the reference method, it is important to note that cholesterol in the fraction it classifies as LDL also includes cholesterol on Lp(a) and some remnant lipoproteins too. In the future, it would, therefore, be important to directly assess the different LDL-C equations, which may be affected differently by cholesterol on these other lipoproteins, for their impact in the clinical management of patients and for their ability to predict future ASCVD events.

  1. IDL-cholesterol is generally quite low compared to LDL-cholesterol in the strict sense. Direct methods (homogeneous methods) show only partial cross-reactivity with IDL. Is there any place for direct methods in the clinical laboratory. These assays exist for more than 20 years but they simply increase confusion.

We share some of the same concerns about direct methods and their limitations but because of space constraints and concern from another reviewer about length of manuscript, we think it is beyond the scope of our current paper to fully discuss this issue.  We did, however, make a short modification to the introduction to briefly state how direct assays have not been fully adopted because of analytical concerns about the specificity of these assays for lipoprotein subfractions raised by the reviewer.  See below: 

Lines 59-62: Because of this issue, which is related to their differential reactivity to different lipoprotein subfractions, and the extra costs for performing direct LDL-C testing, most clinical laboratories in the US still calculate LDL-C, according to recent College of American Pathology proficiency test surveys.

  1. Would it not be more practical to quantify non-HDL cholesterol? Many will argue that remnant cholesterol should be quantified separately. If so, which method should be used? Calculation or method of direct remant cholesterol quantification?

Yes, there are many advocates for just using non-HDL-cholesterol instead of LDL-C.  It is a simple and straightforward calculation but for various reasons it has not been widely adopted.  Most current guidelines still focus on LDL-C and thus the rationale for doing our study, but we have now added this point in the revised manuscript.  See below.  Likewise, as the reviewers points out there are those that also recommend either the measurement or calculation of remnant cholesterol. We did not address this in the revised manuscript because of space constraints but in some of the references that we cite this issue is more fully discussed.

Lines 75-80: For patients with HTG, it is recommended, that a nonHDL-C cut-point of 220 mg/dL, which can be accurately calculated by a simple calculation, be used instead for deciding statin therapy, because of potential inaccuracies in LDL-C estimation [4]. Some have also advocated more widespread use of nonHDL-C as an ASCVD biomarker, but current guidelines still focus most of their recommendations based on LDL-C values. 

Reviewer 2 Report

The authors investigated the accuracy of LDL-cholesterol levels estimating by F-LDL-C, M-LDL-C, eM-LDL-C and S-LDL-C equations compared with the BQ method. However, I have some comments.

1) Did S-LDL-C equation show statistically significant greater accuracy than the other 3 equations?

2) The sections of Results and Discussion seem to be too long and redundant. It can be written much more concisely. 

Author Response

The authors investigated the accuracy of LDL-cholesterol levels estimating by F-LDL-C, M-LDL-C, eM-LDL-C and S-LDL-C equations compared with the BQ method. However, I have some comments.

            We thank the reviewer for carefully reviewing the paper and their useful comments.

  • Did S-LDL-C equation show statistically significant greater accuracy than the other 3 equations?

Yes, we are sorry for this important oversight.  We analyzed the statistical difference between the equations by several different methods for Figures 6 and 7 when we compare potential diagnostic errors that could change the clinical management of patients.  Because of the relatively large sample size, all most all the pairwise comparisons between the different equations showing differences in the number of correctly classified patients was statistically different.  The only exception was when the previously published Martin equation was compared to the enhanced Martin for high TG samples there was no significant difference.  Most importantly and consistent with the main point of our paper, the Sampson equation had statistically less clinically relevant errors than all the other equations.  We have now added a short sentence describing these findings (See below) and have added an additional supplemental table on these new analysis (New Supplemental Table 3).

Lines 107-109: Comparisons among LDL-C equations for the number of potentially clinical relevant errors was done by pairwise Chi-Square analysis and by calculating their kappa scores.

Lines 218-221: In terms of clinically relevant errors (Fig. 6H), a total of 13.5% of the S-LDL-C results would be predicted to potentially change the management of patients, which was statistically less than for F-LDL-C (23.0%), M-LDL-C (20.5%) and eM-LDL-C (20.0%) (Supplemental Table 3).

2) The sections of Results and Discussion seem to be too long and redundant. It can be written much more concisely. 

We have gone over the entire paper and have made several changes to reduce its size, but because it was already below the word limit for this type of article and the request to add more information from the other reviewers, the revised manuscript is only slightly smaller in size.  If the editor would like for us to make some additional changes to further shorten the paper, we will do so in the next version.

Reviewer 3 Report

The paper Biomedicines-2050930 is a validation of the Sampson, Friedwald and Martin equations (used to calculate LDL-cholesterol (LDL-C) from the measured total cholesterol, HDL-cholesterol and triglycerides (TG)) in comparison to directly measured LDL- C. The sample size was big, encompassing a wide range of LDL-C and TG values. The methods are thorough, combining regression analysis, error-grid analysis, and concordance with the direct measurement of LDL-C for different LDL-C ranges. The main result is that the Sampson equation performed best among the tested equations, especially in the high range of TG, as well as in the very low and very high range of directly measured LDL-C.

 The paper is informative and well written. I have only two minor concerns:

-          - The reference (16), describing the authors' original work on the Sampson equation, is incomplete. Please, fill in the missing data.

 -    It would be illustrative to provide the tested equations »side by side« in the Supplementary data, so that the reader can easily comprehend the quadratic term and the intercept of the Sampson equation, as well as the differences between the Friedwald and the Martin equations.

Author Response

The paper Biomedicines-2050930 is a validation of the Sampson, Friedwald and Martin equations (used to calculate LDL-cholesterol (LDL-C) from the measured total cholesterol, HDL-cholesterol and triglycerides (TG)) in comparison to directly measured LDL- C. The sample size was big, encompassing a wide range of LDL-C and TG values. The methods are thorough, combining regression analysis, error-grid analysis, and concordance with the direct measurement of LDL-C for different LDL-C ranges. The main result is that the Sampson equation performed best among the tested equations, especially in the high range of TG, as well as in the very low and very high range of directly measured LDL-C.

 The paper is informative and well written. I have only two minor concerns:

Thank you for the positive comments and for the helpful suggestions.

-                 1. The reference (16), describing the authors' original work on the Sampson equation, is incomplete. Please, fill in the missing data.

Thank you.  We have added the missing information to the reference. 

  1. It would be illustrative to provide the tested equations »side by side« in the Supplementary data, so that the reader can easily comprehend the quadratic term and the intercept of the Sampson equation, as well as the differences between the Friedwald and the Martin equations.

Thanks again for this helpful suggestion.  We have modified the method section by referring to a new supplemental table (Supplemental Table 2) where we now have all the equations side by side as suggested by the reviewer.  See the below changes to the text:

Lines 100-103: LDL-C was calculated by the F-LDL-C [9], M-LDL-C [17], eM-LDL-C [21] and S-LDL-C [16] equations (Supplemental Table 2) by an Excel spreadsheet, which can be downloaded at the following website: https://figshare.com/articles/software/Sampson_LDLC_and_VLDLC_calculator/21346893.

Round 2

Reviewer 2 Report

I have no further comments.